# Single-site decorated copper enables energy- and carbon-efficient CO₂ methanation in acidic conditions

Mengyang Fan [1,4], Rui Kai Miao [1,4], Pengfei Ou[2,4], Yi Xu [1,4], Zih-Yi Lin[3], Tsung-Ju Lee[3], Sung-Fu Hung [3], Ke Xie[2], Jianan Erick Huang[2], Weiyan Ni[2], Jun Li [1], Yong Zhao [1], Adnan Ozden[1], Colin P. O'Brien [1], Yuanjun Chen [2], Yurou Celine Xiao [1], Shijie Liu [1], Joshua Wicks [2], Xue Wang [2], Jehad Abed [2], Erfan Shirzadi[2], Edward H. Sargent [2] ✉ & David Sinton [1] ✉

Renewable CH₄ produced from electrocatalytic CO₂ reduction is viewed as a sustainable and versatile energy carrier, compatible with existing infrastructure. However, conventional alkaline and neutral CO₂-to-CH₄ systems suffer CO₂ loss to carbonates, and recovering the lost CO₂ requires input energy exceeding the heating value of the produced CH₄. Here we pursue CH₄-selective electrocatalysis in acidic conditions via a coordination method, stabilizing free Cu ions by bonding Cu with multidentate donor sites. We find that hexadentate donor sites in ethylenediaminetetraacetic acid enable the chelation of Cu ions, regulating Cu cluster size and forming Cu-N/O single sites that achieve high CH₄ selectivity in acidic conditions. We report a CH₄ Faradaic efficiency of 71% (at 100 mA cm⁻²) with <3% loss in total input CO₂ that results in an overall energy intensity (254 GJ/tonne CH₄), half that of existing electroproduction routes.

Renewable fuels are a critical component of global net-zero emission scenarios and offer high-density long-term energy storage. CO₂ electrochemical reduction (CO₂R) provides a decarbonized path to a variety of chemicals and fuels when powered by renewable electricity[1]. Of the various CO₂R products (mainly carbon monoxide, methane, ethylene, ethanol and n-propanol)[2], methane (CH₄) has the highest energy density of 55.5 GJ/tonne[3], and is a key input for hard-to-decarbonize industries. Renewable synthetic CO₂-derived CH₄ avoids emissions associated with the extraction of fossil-CH₄ (natural gas, NG) and does not add to the natural carbon cycle. As a result, CH₄ produced from captured CO₂ and renewable electricity could provide a pathway to decarbonize existing NG supplies ( > 30% of the world's fossil energy consumption[4,5]), compatible with vast NG storage and distribution infrastructure[6-12].

Present day CO₂R catalysts have achieved Faradaic efficiencies (FEs) of 70-80% towards CH₄ at practical current densities (>100 mA cm⁻²) in alkaline and neutral mediums[13-20]. However, these systems suffer from CO₂ loss to (bi)carbonates, and regenerating the CO₂ is costly[13,14]. In alkaline systems, CO₂ reacts rapidly with excess hydroxides in the electrolytes, consuming 20-fold that reacted productively via CO₂R[21,22]. Recovering the CO₂ requires an energy input of 289 GJₜₕ/tonne CH₄—over 5x the heating value of CH₄ (higher heating value, HHV, 55.5 GJ/tonne, Fig. 1a, c; Supplementary Notes 1–3; Table S1). In neutral CO₂R electrolyzers, CO₂ loss to (bi)carbonates is 4-fold that converted to CH₄. The CO₂ converted to (bi)carbonates (Eq. 1–3), migrates across the anion-selective membrane, combines with the protons from the oxygen evolution reaction (OER) on the anode, reverts to CO₂, and mixes with produced O₂[23,24]. We performed

[1]Department of Mechanical and Industrial Engineering, University of Toronto, 5 King's College Road, Toronto, Ontario M5S 3G8, Canada. [2]Department of Electrical and Computer Engineering, University of Toronto, 10 King's College Road, Toronto, Ontario M5S 3G4, Canada. [3]Department of Applied Chemistry, National Yang Ming Chiao Tung University, Hsinchu, Taiwan. [4]These authors contributed equally: Mengyang Fan, Rui Kai Miao, Pengfei Ou, and Yi Xu. ✉e-mail: ted.sargent@utoronto.ca; sinton@mie.utoronto.ca

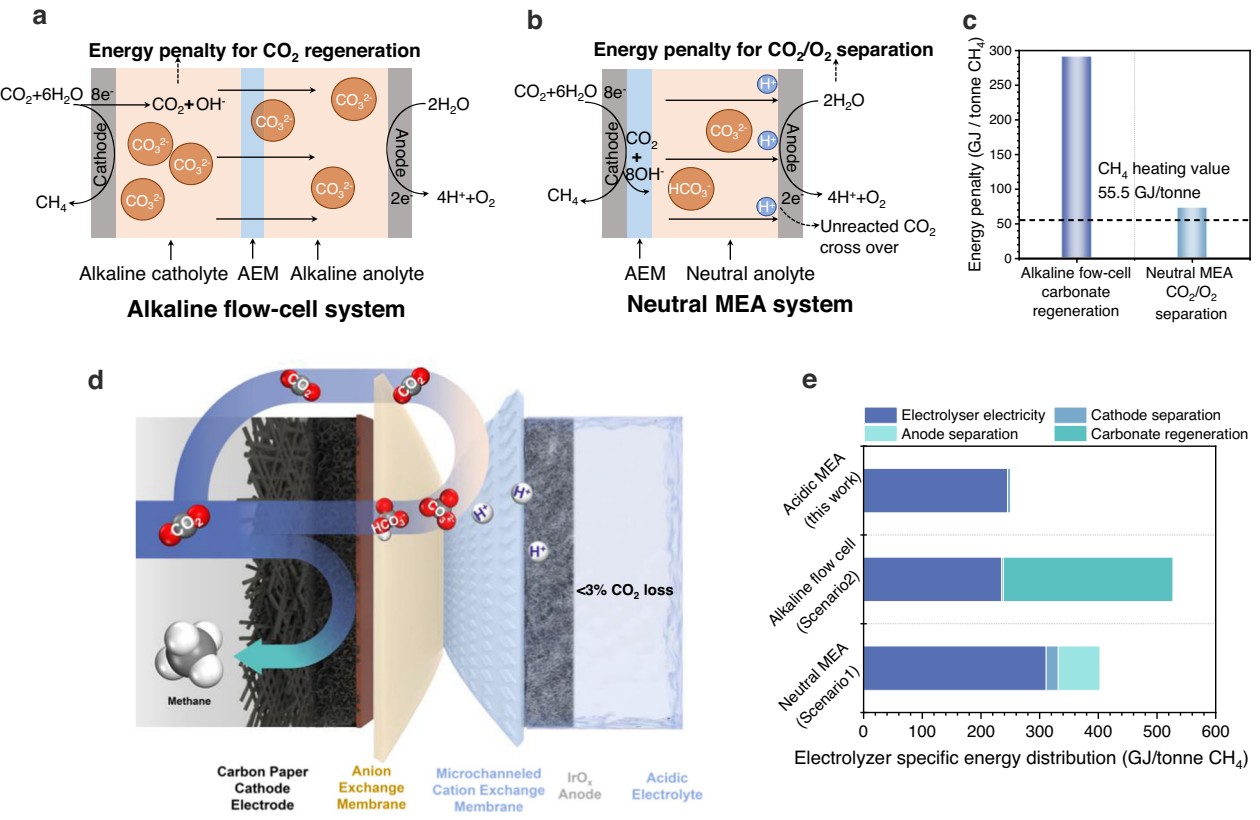

**Fig. 1 | Schematic illustration and techno-energy analysis of different systems.** Schematic figure of (**a**) alkaline flow-cell system and (**b**) neutral MEA system. AEM, anion exchange membrane. MEA, membrane electrode assembly. **c** Energy penalty of $CO_2$ regeneration from carbonate and anodic $CO_2/O_2$ separation. **d** Scheme of the acidic microchanneled MEA system used in this work. **e** Comparison of electrolyzer-specific energy distribution in different systems. Details of the energy analysis are provided in Supplementary Notes 1–4.

neutral-medium $CO_2R$ and found the anode tail gas to consist of 67 v/v % $CO_2$ and 33 v/v % $O_2$ (Supplementary Fig. 1). Separating $CO_2$ from this stream costs 55-73 $GJ_{th}$/tonne $CH_4$, exceeding the $CH_4$ heating value (Fig. 1b, c; Supplementary Notes 4; Table S1). The loss of $CO_2$ fundamentally limits the single-pass conversion (SPC, here defined as the fraction of input $CO_2$ that is reduced to the target product) to <20% in alkaline and neutral electrolyzers[21,25,26]. Achieving high SPC to $CH_4$ will requires carbon efficient systems with minimal $CO_2$ loss[27–29].

$$CO_2 + 6H_2O + 8e^- \rightarrow CH_4 + 8OH^- \qquad (1)$$

$$4CO_2 + 8OH^- \rightarrow 4CO_3^{2-} + 4H_2O \qquad (2)$$

$$8CO_2 + 8OH^- \rightarrow 8HCO_3^- \qquad (3)$$

Here, we demonstrate a $CH_4$-producing membrane electrode assembly (MEA) system that operates in acidic conditions (Fig. 1d). Internal recapture and recycling of $CO_2$, via an internally channeled bipolar membrane, eliminates the need for downstream $CO_2$ regeneration or separation. To enhance $CH_4$ selectivity and minimize hydrogen evolution reaction (HER, Supplementary Fig. 2) in this acidic system, we pursued an in-situ multidentate coordinating strategy, using molecules with multi-teeth as decorations, to constrain Cu(II) from the copper phthalocyanine (CuPc) precursor and regulate Cu cluster size[14,30]. We screened a range of candidates with various multidentate sites as the decorations and found ethylenediaminetetraacetic acid

(EDTA) chelated Cu ions stronger through hexadentate coordination compared with the lower multidentate coordinated molecules. With EDTA decoration, we obtained low-coordinated Cu clusters decorated by Cu-N/O single sites - that facilitate $CO_2R$ to produce $CH_4$. Density functional theory (DFT) computations indicate that these N/O coordinated Cu decoration sites enhance $CH_4$ selectivity by stabilizing the adsorption of *CHO and *O key intermediates. With this strategy we achieve a $CH_4$ FE of 71% at a current density of 100 mA cm$^{-2}$ and a $CH_4$ energy efficiency (EE) of 21%. By eliminating $CO_2$ loss, we achieve a single-pass $CO_2$ conversion of 78%, 5× higher than neutral electrolyzers, and an energy intensity of 254 GJ/tonne $CH_4$ (Fig. 1e). The produced $CH_4$ has 50% the energy intensity of that produced in the best prior electrolyzers.

## Results and discussion
### Carbon-efficient $CO_2$-to-$CH_4$ system optimization
We integrated a cation exchange membrane (CEM) and an anion exchange membrane (AEM) combination in a zero-gap manner as applied previously to achieve high single pass conversion in the generation of multicarbon products[31] (Supplementary Fig. 3). $H_2SO_4$ was employed as the anolyte, providing protons to regenerate $CO_2$ within the MEA cell. We further incorporated various ionomers in the catalyst layer to tune the cathodic local microenvironment (local alkalinity, ion migration and $CO_2$ mass transport)[32–34]. The operating conditions and binder materials were optimized for each case and PiperIon ionomer performed best, with a moderate $CH_4$ FE of 25% and an $H_2$ FE of 45% at a current density of 100 mA cm$^{-2}$ (Supplementary Fig. 4).

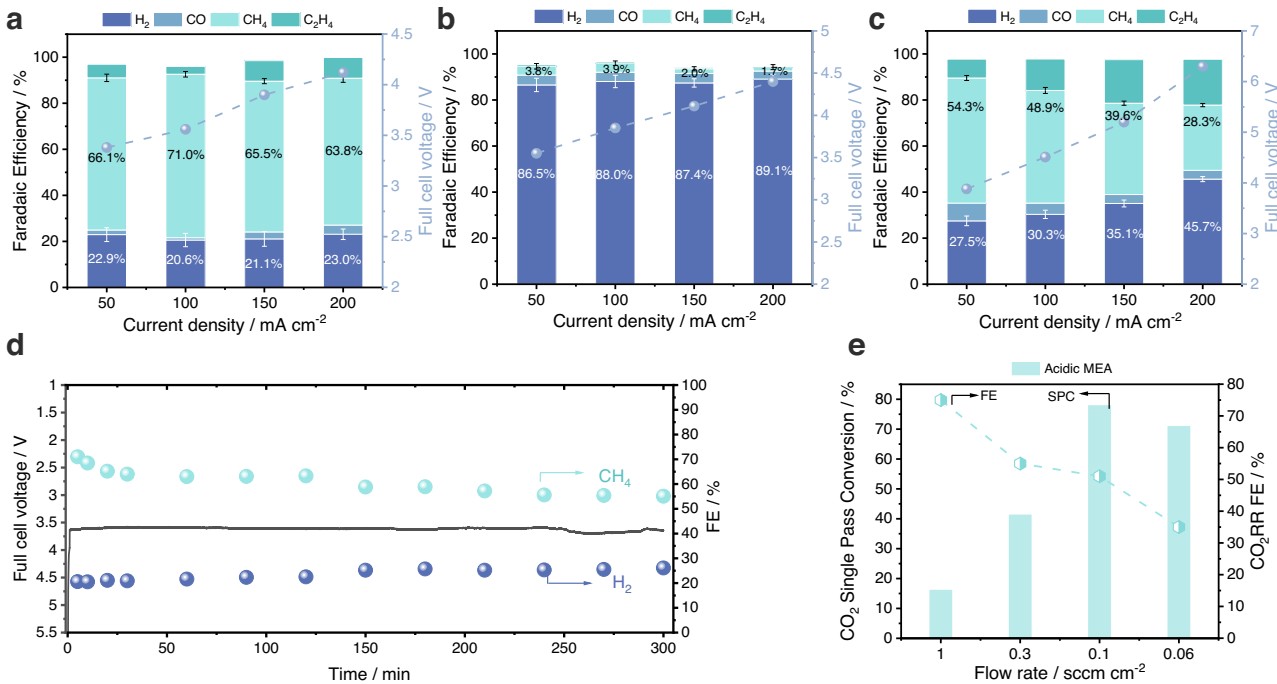

**Fig. 2 | Electrocatalytic performance of electrocatalysts in carbon-efficiency system.** Products distribution of (**a**) EDTA/CuPc/CNP, (**b**) EDTA/CNP and (**c**) EDTA/CuPc at current range from 50 to 200 mA cm⁻². FE, Faradaic Efficiency. Values are means, and error bars indicate SD (*n* = 3 replicates). **d** Stability test of $CO_2$ methanation during 5 h of electrolysis under the current density of 100 mA cm⁻². **e** Single pass conversion of $CO_2$ at different flow rates. The SPC results were obtained at a constant current density of 100 mA cm⁻². SPC, $CO_2$ single pass conversion efficiency.

## Catalyst performance

To enhance the selectivity of $CH_4$, we deployed the low-coordination Cu strategy that is selective for $CO_2$ electrochemical methanation[14,30]. Low-coordinated Cu sites can be produced from the in-situ reduction of Cu(II)Pc precursors during $CO_2R$ and using conductive carbon nanoparticles (CNP) as modulators, confining the Cu cluster size[14]. Without the constraining effect of CNP, free Cu ions readily agglomerates into large Cu clusters, forming high-coordination number sites[14,35] that shift the reaction from $CO_2$ hydrogenation to C-C coupling[14,17,36]. CNP can sterically distribute the metallic Cu clusters and restrict the Cu agglomeration size, which is essential to preserve $CO_2$ hydrogenation activity[14]. Varying the CuPc/CNP ratio from 5:1 to 1.5:1, $CH_4$ FE increased from 21% to 46%. Further increasing CNP content elevated $H_2$ FE (Supplementary Fig. 5). Within the initial hour of the electrolysis, we observed a rapid decay of $CH_4$ FE (from 46% to 25%) accompanied by an increase in $C_2H_4$ FE (from 8% to 19%), which we attributed to the continuous leaching of Cu ions and agglomeration in this acid system (Supplementary Fig. 6)[37,38].

To further increase $CH_4$ selectivity, we designed a multidentate chelating strategy that captures and constrain free Cu ions[14,39–41]. We screened several typical molecules that enable bonding Cu ions through multidentate donor sites. EDTA presents a stronger chelating effect on Cu through hexadentate coordination compared with the lower multidentate coordinated complexes (Supplementary Fig. 7). We fabricated the molecule decorated CuPc/CNP composite catalysts by spray-coating the mixture onto the gas diffusion layers (GDLs). After the initial hour of electrolysis, the ethylenediamine (ED, bidentate coordinated with Cu) and ethylenediamine-N, N′-diacetic acid (EDDA, tetradentate coordinated with Cu) decorated samples showed lower ethylene FE (17% and 11% respectively) than the sample without decorations (19%). The $CH_4$ FE was 32% for the ED decorated sample and 41% for EDDA decorated sample, slightly improved over the no-decoration case. For the EDTA decorated sample, $CH_4$ FE remained >65% after the initial hour of electrolysis with minimal increase in $C_2H_4$

FE (5%, Supplementary Fig. 8). We attribute this improvement in $CH_4$ production to the hexadentate coordinated sites of EDTA that more intensely chelate Cu ions than the other two complexes (Supplementary Fig. 9).

The EDTA loading was screened with a fixed CuPc/CNP ratio of 1.5:1 (Supplementary Fig. 10). At 100 mA cm⁻², the EDTA/CuPc/CNP attained a $CH_4$ FE of 71%, a 20% improvement over the CuPc/CNP case (Fig. 2a). The $CH_4$ FE of > 60% was maintained over a wide current window from 50 to 200 mA cm⁻² with full cell potentials <4.2 V. The $CH_4$ FE remained constant during the initial hour (Supplementary Fig. 11), indicating the regulation of Cu ions by the multidentate chelation effect. The control sample EDTA/CNP showed only a trace $CH_4$ FE of ~4% and an $H_2$ FE of ~90% (Fig. 2b), indicating that in the absence of Cu sites, EDTA is not an active catalyst for $CO_2$-to-$CH_4$ conversion. Without CNP, EDTA-decorated CuPc (0.1 mg/cm²) showed a $CH_4$ FE of 48.9% (Fig. 2c), 20% higher than the pristine CuPc electrode with the same loading (Supplementary Fig. 12). These results evidence the critical role of multidentate chelating effect of EDTA in enhancing $CH_4$ FE. However, in the absence of CNP conductors and Cu modulators, the full cell voltage was high ( > 6 V at 200 mA cm⁻², Fig. 2c). The $CH_4$ selectivity also suffered without CNP regulators and production shifted to $C_2H_4$ FE ($CH_4$:$C_2H_4$ shifted from 20:1 to 4:1). The FE of liquid products were quantified, and the total measured FE approached 100% at the same current density range (50 to 200 mA cm⁻²) in all three cases, within experimental error. (Supplementary Fig. 13). Control experiments were carried out under Ar conditions to rule out EDTA and CNP as the potential carbon sources in the production of carbon-based products. The exclusive $H_2$ production under such conditions indicates that EDTA and CNP were not reactive carbon sources (Supplementary Fig. 14).

We performed a durability test of EDTA/CuPc/CNP in the acidic MEA with 5-mM $H_2SO_4$ anolyte. The $CH_4$ FE remained over 50% with a steady full cell potential of 3.6 V for 5 h (Fig. 2d). We compared the $CO_2R$ performances (Supplementary Fig. 15) and $CO_2$ single pass

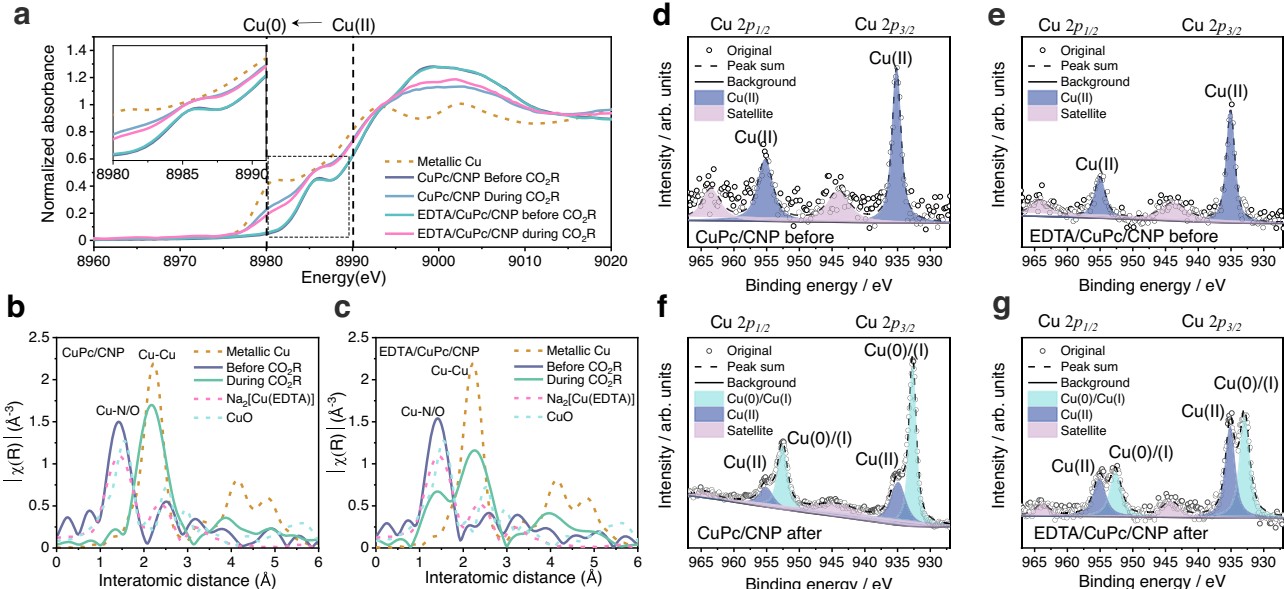

**Fig. 3 | Surface characterizations of catalysts. a** The in-situ XANES spectra of CuPc/CNP and EDTA/CuPc/CNP samples before and during $CO_2R$. The metallic Cu was shown as reference. XANES, X-ray absorption near-edge structure. The in-situ EXAFS spectra of (**b**) CuPc/CNP and (**c**) EDTA/CuPc/CNP before and during $CO_2R$. The metallic Cu, CuO, and $Na_2[Cu(EDTA)]$ samples are shown as references. EXAFS, extended X-ray absorption fine structure. The in-situ experiments were all performed at 100 mA $cm^{-2}$, the current condition for the best $CO_2R$ performance. XPS Cu 2p spectra of (**d**) CuPc/CNP, and (**e**) EDTA/CuPc/CNP before $CO_2R$. XPS Cu 2p spectra of (**f**) CuPc/CNP, and (**g**)EDTA/CuPc/CNP after $CO_2R$. XPS, X-ray photoelectron spectra.

conversion (SPC) in our acidic system with the conventional neutral (0.5 M $KHCO_3$ anolyte) and alkaline systems (0.5 M KOH anolyte). With a total $CO_2R$ FE > 50%, the total $CO_2$ SPC achieved 78%, 4-fold the theoretical maximum of 20% for neutral and alkaline systems (Fig. 2e, Supplementary Fig. 16). The SPC towards $CH_4$ attained a record-high value of 51%, 3.6-fold higher that of neutral medium electrolyzers with the same catalyst (14%, Supplementary Fig. 17). In neutral/alkaline systems, the (bi)carbonates cross through the AEM, leading to the $CO_2$ loss. The CEM in the acidic MEA provided a locally acidic domain for $CO_2$ regeneration within the cell and thereby minimized $CO_2$ loss (< 3 v/v % $CO_2$ detected in the anode tail gas, Supplementary Fig. 1) and achieved high $CO_2$ single pass conversion. The CEM and the integrated microchannels do not add significant ohmic resistance to the overall system[31], as indicated by the comparable voltage with the neutral system (Supplementary Fig. 15).

## Surface characterizations of catalysts

To investigate the multidentate chelating effect and probe the mechanism behind the enhanced $CH_4$ selectivity, we investigated the electronic state and coordination number of Cu at a current density of 100 mA $cm^{-2}$ with in-situ X-ray absorption spectroscopy (XAS). The CuPc/CNP samples were analyzed, with and without EDTA, and metallic Cu, CuO, $Na_2[Cu(EDTA)]$ samples were taken as references. We found from the X-ray absorption near-edge structure (XANES) spectra that both with and without EDTA decoration the original Cu(II) peaks present at 8991 eV. These Cu(II) original peaks shifted to 8980 eV during $CO_2R$, indicating the electronic state of Cu reduced from Cu(II) to the lower state of Cu(0) (Fig. 3a), as expected for Cu clusters were formed by CuPc reduction during $CO_2R$[14]. The EDTA decorated CuPc/CNP sample showed a slightly higher energy position between 8980 ev and 8991 eV (compared to the bare CuPc/CNP sample decoration), indicating the preservation of oxidized states of Cu species during $CO_2R$ (Fig. 3a)[42,43].

We then obtained the in-situ extended X-ray absorption fine structure (EXAFS) spectra to investigate the Cu coordination environments. The sample without decoration showed an increase in Cu-Cu peak in the initial hour, and a sharp drop in Cu-N/O peak (coordination number dropped from 3.8 to 0.6) during the $CO_2R$ process, indicating the Cu agglomeration (Fig. 3b and supplementary Fig. 18, 19, Table S2). This Cu agglomeration leads to decline of $CH_4$ selectivity during $CO_2R$ (Supplementary Fig. 6)[30]. With EDTA decoration, the Cu-Cu peak increased and Cu-N/O peak declined in the initial 30 min, then remained stable for the rest of the process (Supplementary Fig. 20), demonstrating the regulation of Cu ions via the chelating effect (Fig. 3c). We obtained small Cu clusters decorated with additional Cu-N/O sites. The fitted Cu-Cu bond coordination number of the EDTA decorated sample is smaller (5.4) than that of pristine CuPc/CNP (6.7), demonstrating the multidentate chelation constraining effect on Cu cluster size (Supplementary Table S2) The fitted Cu-N/O coordination number of the EDTA decorated sample was larger (2.5, Supplementary Fig. 21, Table S2) than the sample without EDTA decoration (0.6). We attributed the enhanced and maintained $CH_4$ FE (Supplementary Fig. 11) to the EDTA chelating effect on Cu ions -that confined Cu cluster size and generated additional Cu-N/O active sites[13,35].

To further investigate the catalyst surface structures, we performed X-ray photoelectron spectra (XPS) and scanning electron microscope (SEM) before and after $CO_2$ electrolysis. All samples were processed in a glove box after $CO_2R$ to protect them from being oxidized in the air. The Cu 2p XPS spectra of both samples before $CO_2R$, with and without EDTA decoration, showed the peaks at 955.0 eV (2p/1/2) and 935.1 eV (2p3/2), which are associated with the Cu(II) state (Fig. 3d, e). For the post-electrolysis samples, the deconvoluted Cu 2p spectra show Cu(0)/(I) peaks located at 944.4 (2p/1/2) and 932.9 eV (2p3/2), further confirming that Cu(0) sites were formed in the $CO_2R$ process (Fig. 3f, g). However, the EDTA decorated sample presented a smaller Cu(0)/(I) peak ratio than the sample without decoration. We quantified the Cu(0)/(I):Cu(II) peak ratio by integrating the peak area for both Cu(0)/(I) and Cu(II). The EDTA decorated sample showed lower Cu(0)/(I):Cu(II) peak ratios compared to the sample without decoration (Supplementary Table S3), indicating that multidentate chelating decoration is essential to regulation of Cu ions—a finding consistent with the in-situ XAS results. The deconvoluted N 1 s spectra

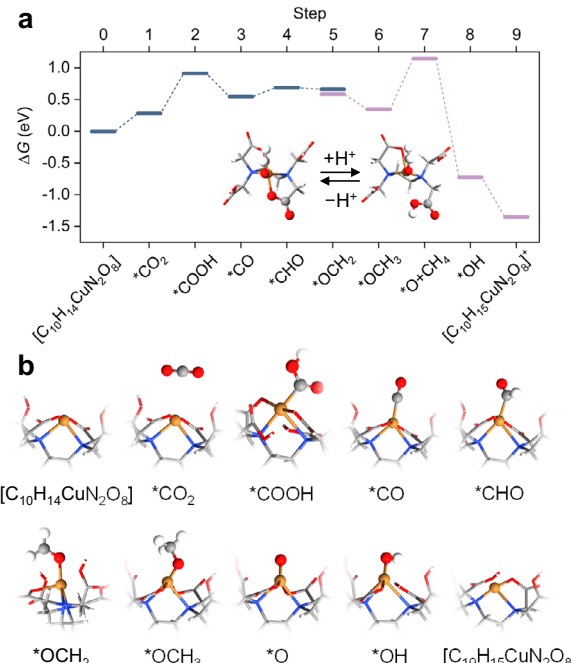

**Fig. 4 | DFT calculations on $CO_2$ protonation to $CH_4$. a** Free energy diagram for $CH_4$ production on Cu active site in the complex structures of $[C_{10}H_{14}CuN_2O_8]$ and $[C_{10}H_{15}CuN_2O_8]^+$. The inserted figures represent the protonation/deprotonation between $[C_{10}H_{14}CuN_2O_8]$ and $[C_{10}H_{15}CuN_2O_8]^+$. **b** Corresponding atomic configurations for each elementary step, including $[C_{10}H_{14}CuN_2O_8]$, *$CO_2$, *COOH, *CO, *CHO, *$OCH_2$, *$OCH_3$, *O, *OH, and $[C_{10}H_{15}CuN_2O_8]^+$. Orange, red, gray, white, blue sphere represent Cu, O, C, H, N atoms, respectively.

also showed that the Cu-N bond was decomposed in the post-electrolysis states of both samples (Supplementary Fig. 22, 23). To further prove the multidentate chelating effect on the stabilization of the Cu-N bond, we normalized the Cu-N bond with reference to inert pyridinic N to calculate the Cu-N loss ratio (Supplementary Table S4). The sample with EDTA decoration demonstrated a lower Cu-N loss (35%) compared to the one without decoration (57%), consistent with the in-situ XAS results, further confirming the formation and preservation of Cu-N sites facilitated by multidentate coordination of EDTA. We then assessed catalyst structure change before and after $CO_2R$ using SEM (Supplementary Fig. 24, 25). The sample without any decoration presented a needle-like structure attributed to the unconstrained Cu deposition when reducing from CuPc during $CO_2R$. In contrast, the EDTA decorated sample was composed of uniformly distributed particles, attributed here to effect of the multidentate coordination in preventing non-regulated Cu deposition during $CO_2R$.

## DFT calculations
To further probe the effect of the decorated Cu-N/O sites on boosting the $CH_4$ selectivity, we performed DFT calculations on a series of $[C_{10}H_{14+/-n}CuN_2O_8]^{n+/-}$ ($n = 0$, 1, or 2) complex structures. We presented the free energy diagram of the lowest-energy pathway for $CO_2$-to-$CH_4$ on Cu active site in the complex structures of $[C_{10}H_{14}CuN_2O_8]$ and $[C_{10}H_{15}CuN_2O_8]^+$ (Fig. 4a) and the corresponding atomic configurations of each elementary step (Fig. 4b). The $CH_4$ production initiates from thermodynamically inhibited adsorption and protonation of $CO_2$ on $[C_{10}H_{14}CuN_2O_8]$ (Fig. 4a), similar to the results on Cu(111) facet[44], with free energy changes of 0.28 eV and 0.63 eV, respectively. The potential-determining step (PDS) is the protonation of *$OCH_3$ to *O + $CH_4$, exhibiting a free energy change of 0.80 eV. We noted that the protonation/deprotonation between $[C_{10}H_{14}CuN_2O_8]$ and $[C_{10}H_{15}CuN_2O_8]^+$ (Fig. 4a, inset) is a thermal-neutral step (with a free energy of 0.08 eV for Step 5 and −0.02 eV for Step 9 to Step 0). Such a

configuration of $[C_{10}H_{14}CuN_2O_8]$ and $[C_{10}H_{15}CuN_2O_8]^+$ can stabilize the adsorption of *CHO and *O, which decreases the free energy needed for the *CO protonation to *CHO and *$OCH_3$ protonation to *O. The PDS on Cu(111) facet is the protonation of *CO species (i.e., *CO-to-*CHO) with a free energy change of 0.85 eV. Compared to Cu(111) facet, we found that $[C_{10}H_{14}CuN_2O_8]$ enables $CO_2$ adsorption and exhibits a comparable free energy change for the PDS, indicating that $[C_{10}H_{14}CuN_2O_8]$ offers extra active sites for $CO_2$-to-$CH_4$ process. The formation of $[C_{10}H_{14}CuN_2O_8]$ sites also prevents the agglomeration of Cu clusters, lowers the *CO coverage on the Cu(111) facet, and inhibits C-C coupling – collectively enhancing $CH_4$ selectivity.

In summary, this work presented a $CH_4$-selective single-site decorated Cu strategy compatible with a carbon-efficient system. Employing acidic conditions in a structured MEA electrolyzer eliminated $CO_2$ loss and the associated energy cost of $CO_2$ regeneration. We developed an multidentate chelating strategy to obtain Cu-N/O single sites decorated low-coordinated Cu that enables 71% FE of $CH_4$ in this carbon-efficient system. We obtained a full cell potential of 3.6 V at 100 mA cm⁻² and a record-high SPC towards the $CH_4$ of 51% and an energy efficiency for $CH_4$ production of 21%. By avoiding the additional energy consumption of $CO_2$ regeneration and improving the energy efficiency of $CO_2$-to-$CH_4$, renewable $CO_2$-derived $CH_4$ is produced at an overall energy cost of ~254 GJ/tonne−50% less than the conventional alkaline and neutral approaches. This study demonstrates a strategy to simultaneously achieve carbon- and energy-efficient $CO_2$ methanation.

## Methods
### Preparation of electrodes
The electrodes were prepared by air-bushing the CuPc/CNP or EDTA/CuPc/CNP inks onto hydrophobic carbon papers. The spray density was kept at 0.1 mL cm⁻². The CuPc/CNP catalyst inks were prepared by dispersing 64 mg CuPc ( > 99.5%, Sigma-Aldrich) and 42 mg CNP (Vulcan XC 72, Fuel Cell Store) mixture in 30 mL methanol (>99.5%, Fisher chemical) with 150 μL 5 wt% PiperIon (Fuel Cell Store) anion exchange ionomer as the binder. We obtained different CuPc/CNP ratios by varying the CNP quantity in the mixture, and the CuPc/CNP ratio was ranged from 1:4 to 5:1. The molecule (ED, EDDA and EDTA) decorated CuPc/CNP inks were prepared by adding 16 mg decoration molecule (ED > 99.5% Sigma-Aldrich; EDDA > 98% Sigma-Aldrich; EDTA > 99.5%, Sigma-Aldrich) into 2 mL fully dispersed CuPc/CNP solutions, followed by sonicating for 24 h. The EDTA and CuPc ratio were optimized from 16:1 to 2:1 by tuning the EDTA weight in the CuPc/CNP dispersed solution.

### Acidic MEA configuration
The cathodes for the acidic MEA were based on either CuPc/CNP or EDTA decorated CuPc/CNP electrodes with the catalyst loading of 0.15 mg cm⁻². The anodes were based on Ti felt (0.3 mm thickness) loaded with 1 mg cm⁻² $IrO_2$. 0.005 M $H_2SO_4$ was used as anolyte circulated with a flow rate of 5 mL min⁻¹. A microchanneled cation exchange membrane (Nafion 117, Fuel Cell Store) facing the anodic side was used for transporting proton and locally regenerated $CO_2$. The channeled CEM was prepared by hot embossing under a temperature of 220 °F and a pressure of 1.25 MPa for 5min[31]. An anion exchange membrane (Sustainion X37-50 Grade RT, Dioxide Materials, USA) facing the cathodic side was used to facilitate $CO_2R$ activation. DI water was circulated in the middle channel layer at a constant flow rate of 0.5 mL min⁻² using a syringe pump.

### Electroreduction of $CO_2$
The $CO_2R$ was carried out at constant current densities ranging from 50 to 200 mA cm⁻². The gas products were analyzed in 1 mL volume through a gas chromatograph (GC, Perkin Elmer Clarus 590) equipped with a thermal conductivity detector (TCD) and a flame ionization detector (FID). The Faradaic efficiency was calculated via the following

equation:

$$Faradaic\ efficiency\,(\%) = \frac{zFP}{RT} \times v \times \frac{1}{I} \times 100\% \qquad (4)$$

where $z$ represents the number of electrons required to produce the product, $F$ represents the Faraday constant, $P$ represents the atmosphere pressure, $R$ represents the ideal gas constant, $T$ represents the temperature, $v$ represents the gas flow rate at the gas, and $I$ represents the total current.

The full cell voltage was obtained during $CO_2R$ and the energy efficiency was calculated using the following equation:

$$Energy\ efficiency\,(\%) = \frac{E_i^0}{E_{cell}} \times FE \times 100\% \qquad (5)$$

where $E_i^0$ is the thermodynamic potential, $E_{cell}$ is the full cell potential voltage during the experiments, and FE is the Faradaic efficiency of each product.

The single-pass $CO_2$ conversion efficiency (SPC) of $CO_2$ was calculated using the following equation:[45]

$$SPC\,(\%) = \frac{\frac{j}{zF} \times V_m}{flow\ rate} \times 100\% \qquad (6)$$

where $j$ represents the partial current density of a specific product, $z$ represents the number of electrons required for the specific product, $F$ represents the Faraday constant, $V_m$ represents the molar volume.

## Characterizations of catalysts

Cu catalyst electronic state and the local coordination environment were investigated by in-situ XAS measurements, which were performed at beamline 9BM of the Advanced Photon Source (APS, Argonne National Laboratory, Lemont, Illinois, United States) and the silicon drift detector at the 17 C beamline of National Synchrotron Radiation Research Center (NSRRC, Hsinchu, Taiwan)[18]. The ex-situ XPS spectra were obtained through a Thermo Scientific K-Alpha spectrophotometer with the monochromated Al Kα X-ray radiation source. The ex-situ samples were treated and stored strictly under the $N_2$ condition to reduce the possible oxidation of Cu. SEM characterizations were conducted with a high-resolution scanning electron microscope (HR-SEM, Hitachi S-5200).

## DFT calculations

First-principles calculations based on DFT[46,47] were performed using the projector-augmented wave method[48,49] as implemented in the Vienna ab initio simulation package (VASP). Electron exchange and correlation terms were treated[50] by generalized gradient approximation which is parametrized by Perdew-Burke-Ernzerhof with long-range dispersion correction derived from the DFT-D2 method of Grimme[51]. $[C_{10}H_{14+/-n}CuN_2O_8]^{n+/-}$ ($n = 0$, 1, or 2) was modeled in a supercell with a vacuum thickness >20 Å in each direction. Cut-off energy was set to 450 eV and the Brillouin zone was sampled by gamma-centered $1 \times 1 \times 1$ $k$-points generated by the Monkhorst-Pack scheme[52]. Structural optimization was considered to reach the convergence when the residual force on each ion was <0.01 eV Å$^{-1}$ and the energy difference between the two iterations was <$10^{-5}$ eV per atom. A Fermi-level smearing width of 0.05 eV was used for the calculations of adsorbates, whereas 0.01 eV for non-adsorbed species, to improve the convergence.

## Data availability

Data that support the findings of this study can be found in the article and the Supplementary information. Source data are available from the corresponding author upon request.

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

## Acknowledgements

The authors acknowledge support from the Natural Sciences and Engineering Research Council (NSERC) of Canada, Natural Resources Canada—Clean Growth Program, and the Natural Gas Innovation Fund (NGIF). The infrastructure provided through the Canada Foundation for Innovation and the Ontario Research Fund supported the work. R.K.M. thanks NSERC, Hatch, and the Government of Ontario for their support through graduate scholarships. P.O. thanks the Climate Positive Energy for its support through Rising Stars in Clean Energy Postdoctoral Fellowship. Synchrotron experiments were carried out at SXRMB beamline at the Canadian Light Source (CLS). S.F.H. thanks for the supports from the National Science and Technology Council, Taiwan (Contract No. NSTC 111–2628-M-A49-007) and the support from the Yushan Young Scholar Program, Ministry of Education, Taiwan. W.N. acknowledges the financial support of the Postdoc. Mobility Fellowship from the Swiss National Science Foundation (SNSF) (No. P500PN_202906). We acknowledge the help from Dr. Qunfeng Xiao, Dr. Mohsen Shakouri, and Dr. Alisa Paterson for their kind technical assistance.

## Author contributions

D.S. and E.H.S. supervised the project. M.F. and Y.X. designed and carried out all the experiments. M.F. designed the catalysts and performed catalysts characterizations. R.K.M., A.O., and J.W. carried out the TEA analysis. P.O. carried out the DFT simulation. M.F., R.K.M, and K.X. analyzed the experimental data and prepared the manuscript. R.K.M. performed the SPC experiments. Y.C. synthesized the catalysts and Y.Z. carried out SEM characterizations, Z.-Y.L., T.-J.L., W.N., J.A., and E.S. carried out XAS characterization. S.-F.H., J.L., and X.W. analysed the XAS data. J.E.H., C.P.O., S.L., and Y.C.X. performed product analysis. All authors discussed the results and assisted during manuscript preparation.

## Competing interests

The authors declare no competing interests.
