## [Peer review file · Nature Communications]

REVIEWER COMMENTS

Reviewer #1 (Remarks to the Author):

In this work, the authors presented a CH₄-selective single-site decorated Cu strategy compatible with a carbon-efficient system. Hexadentate donor sites in ethylenediaminetetraacetic acid (EDTA) enable the stabilization of Cu(II) by forming Cu-N/O single sites that achieve high CH₄ selectivity in acidic conditions. The result shows a full cell potential of 3.6 V at 100 mA cm⁻² and a record-high SPC towards the CH₄ of 51% and an energy efficiency for CH₄ production of 21%. Some relevant characterization techniques and catalytic experiments needed to initially understand the system were included. The conclusions are basically supported by experimental and theoretical results. However, there are some questions should be solved. Thus, I will recommend its publication in Nature Communications after a major revision.

Some specific comments are below:

1. The author claim that they developed an multidentate chelating strategy to obtain Cu-N/O single sites decorated Cu to enable the stabilization of Cu(II). However, the in-situ Cu EXAFS spectra in Figure S17 show that the Cu-N/O peak and Cu-Cu peak is unstable in 30 min, 60 min, 90 min. It seems that the Cu(II) and Cu of EDTA/CuPc/CNP are changed in the electrolysis process. Please explain it. In addition, the stability test in Figure 2d show that the FE decline constantly in the electrolysis. What's the reason for the degradation of catalytic activity in the electrolysis? What are the real catalytic site, Cu(II) or Cu?
2. In the process of electrochemical CO₂RR, PH is an important factor. The CO₂RR performance comparison of CuPc in acidic MEA and neutral MEA system also proves it in this work. The result show that the CO₂RR performance is better in neutral MEA than that the acidic MEA system. The author should provide the CO₂RR performance of EDTA/CuPc/CNP in neutral and alkali MEA system to show the superiority of acidic MEA system and explain the reason of this phenomenon.
3. It seems that all the total FE is lower than the unity (i.e., 100%), the authors should discuss this crucial point.
4. An isotopic experiment using ¹³CO₂ as substrate should be performed in order to prove the origin of CH₄, C₂H₄ and CO and guarantee its formation quantity.

Reviewer #2 (Remarks to the Author):

In this manuscript, Fan et al. reported an EDTA/CuPc/CNP hybrid electrocatalyst for CO₂ reduction to methane with both high energy and carbon efficiency. Such catalyst could exhibit CH₄ Faradaic efficiency (FE) up to 71% at 100 mA cm⁻² with <3% loss in total input CO₂. The good performance was attributed to the hexadentate donor sites in EDTA, enabling the stabilization of Cu(II) by forming Cu-N/O single sites. Overall, it is an interesting study, but there are several technical problems, which need to be addressed before considering its publication.

1), The important innovation in this paper is the high single-pass CO₂ conversion efficiency of 78%. It should be noted that such high CO₂ utilization efficiency was mainly due to its operation in a zero-gap MEA comminating a cation exchange membrane (CEM) and an anion exchange membrane (AEM), which has been reported in their previous work (*Joule* 2022, 6 (6), 1333–1343). The result is not surprising. It should no longer be used as an innovative point in this article and spend a large space to illustrate.

2), The author emphasizes severer times that the addition of EDTA helps to suppress HER. In fact, the inhibition of H₂ mainly comes from the choice of binder and the addition of CNP. For example, by replacing the Nafion binder with Piperion, FE(H₂) dropped from 67% to 47%. Then it can farther be reduced to 20% with only CNT. The detailed role of EDTA for suppressing HER should be described.

3), The authors believe that the formation of single Cu-O/N sites decorated low-coordinated Cu clusters is crucial for producing methane efficiently. They observed the existence of Cu-O/N sites with in-situ EXAES, but there was no direct evidence that it was the key catalytic site. Considering previous studies that Cu clusters were also efficient in producing methane, it is confusing whether the key active site is Cu-O/N site or low-coordinated Cu cluster.

4), High CH₄ selectivity has also been reported by adding chelating ligand molecules directly to the surface of polycrystalline copper (EDTMPA on commercial polycrystalline Cu, *Nat. Commun.* 2022, 13 (1), 3158.) . The authors need to compare previous reports to illustrate the major innovation of this work.

5), In the theoretical calculation section in figure 4, why the charge of the catalyst in step 9 is not the same as the one in step 0?

6), The authors should discuss the expense of their system with lower energy intensity, such as the use of both AEM and CEM, microfluid channels. I am also curious on the stability of the system operated at higher current densities.

Reviewer #3 (Remarks to the Author):

In this manuscript, Fan et al. report a high selectivity (Faradaic efficiency of 71%) for the conversion of CO₂ to methane under acidic conditions on a dispersed Cu electrocatalyst at practical current densities (~100 mA/cm²). Importantly, they demonstrate a high single-pass conversion of CO₂ to this product under minimal loss of CO₂ to bicarbonate.

This work is of significance because the energy efficiency of the conversion of CO₂ under alkaline and neutral conditions is severely limited by the enviable loss of a significant fraction of CO₂ to bicarbonate under these conditions. Selective CO₂ reduction under acidic conditions is challenging because of the competition with the hydrogen evolution reaction. The authors demonstrate a clever strategy for stabilizing the undercoordinated Cu sites that promote CO₂-to-methane conversion.

The manuscript is well written, the data appears to be robust and supports the authors' claims. Methods and materials are described in appropriate detail. A strength of the work is the combination of technoeconomic analysis, catalyst design, careful reactor engineering, and catalyst characterization. Few articles approach a subject in such a comprehensive manner. The authors have thoroughly characterized their catalytic system with X-ray spectroscopy (Fig. 3) and DFT (Fig. 4). The characterization reveals no surprising insights (the mechanistic aspects have been mostly established in prior work and the active site structures on Cu have also been extensively described in the literature). Therefore, I estimate that the primary impact of this work on the field will be that it represents a new benchmark for CO₂-to-methane conversion under acidic conditions.

I did not find any technical weaknesses or issues with the presentation. The manuscript is publishable in its present form.

AUTHORS' REPLY TO THE REVIEWERS' COMMENTS

Title: Single-site Decorated Copper Enables Energy- and Carbon-efficient Electroproduction of Synthetic Methane

Journal: Nature Communications (MS ID: NCOMMS-23-03695-T)

We thank all reviewers for their review of this work and for helpful recommendations.

Reviewer #1 (Comments for the Author):

In this work, the authors presented a CH₄-selective single-site decorated Cu strategy compatible with a carbon-efficient system. Hexadentate donor sites in ethylenediaminetetraacetic acid (EDTA) enable the stabilization of Cu(II) by forming Cu-N/O single sites that achieve high CH₄ selectivity in acidic conditions. The result shows a full cell potential of 3.6 V at 100 mA cm⁻² and a record-high SPC towards the CH₄ of 51% and an energy efficiency for CH₄ production of 21%. Some relevant characterization techniques and catalytic experiments needed to initially understand the system were included. The conclusions are basically supported by experimental and theoretical results. However, there are some questions should be solved. Thus, I will recommend its publication in Nature Communications after a major revision.

Response: We thank the Reviewer #1 for the detailed comments and have revised the manuscript accordingly as detailed in-line below.

Specific Comments:

Comment #1:

The author claim that they developed an multidentate chelating strategy to obtain Cu-N/O single sites decorated Cu to enable the stabilization of Cu(II). However, the in-situ Cu EXAFS spectra in Figure S17 show that the Cu-N/O peak and Cu-Cu peak is unstable in 30 min, 60 min, 90 min. It seems that the Cu(II) and Cu of EDTA/CuPc/CNP are changed in the electrolysis process. Please explain it. In addition, the stability test in Figure 2d show that the FE decline constantly in the electrolysis. What's the reason for the degradation of catalytic activity in the electrolysis? What are the real catalytic site, Cu(II) or Cu?

Response #1:

We have re-examined the in-situ Cu EXAFS spectra and updated our Supplementary Fig. 18 and 20. The pristine CuPc/CNP sample showed a continuous increase in Cu-Cu peak in 30 and 60 min and a sharp decrease in Cu-N/O peak when a reduction current was applied (new **Supplementary Fig. 18**). The Cu-N/O coordination number is 0.6, much less than the value of 3.8 corresponding to the pre-electrolysis state (**Supplementary Table 2**). We attribute this sharp loss of Cu-N/O to the leaching of Cu(II) out of CuPc during CO₂R. The Cu-Cu peak increases during the initial 30

to 60min of CO₂R, indicating subsequent agglomeration of Cu (new **Supplementary Fig. 18**). This Cu agglomeration leads to decline of CH₄ selectivity during CO₂R (**Supplementary Fig. 6**)

By incorporating EDTA decoration, Cu-Cu peak increases in the initial 30 min but is stable for the remaining 120 min (new **Supplementary Fig. 20**). In addition, there is only a small decline in Cu-N/O peak in the first 30 min, attributing to the initial Cu(II) leaching. The EDTA chelated the free Cu ions through N/O bonds, regulating Cu cluster size. By using the chelating strategy, we obtained small Cu clusters decorated with additional Cu-N/O sites. The fitted Cu-Cu coordination number is smaller (5.4) compared with the pristine CuPc/CNP (6.7) sample without EDTA decoration. The Cu-N/O coordination number of EDTA case is 2.5, larger than the 0.6 of the case without EDTA. We attributed the enhanced and maintained CH₄ FE (**Supplementary Fig. 11**) to the EDTA chelation effect on Cu ions, which confined Cu cluster size and generated additional Cu-N/O active sites.

The degradation shown in the original Fig. 2d was due to the operation condition. We used a syringe pump to circulate DI water within the channel layer at a flowrate that was too high (1 mL min⁻¹), and the excess water altered the catalyst local environment. We have performed a new stability test with a lower water circulation rate (0.5 mL cm⁻²). In the revised **Fig. 2d**, the CH₄ FE remains stable at 55% during 5h electrolysis.

Action #1:

- On Page 25 and 27 of the revised supplementary information, we updated the Supplementary Fig. 18 and 20.
- On Page 7, Lines 158-173 of the revised manuscript, we modified our description as:

“We then obtained the *in-situ* extended X-ray absorption fine structure (EXAFS) spectra to investigate the Cu coordination environments. The sample without decoration showed an increase in Cu-Cu peak in the initial hour, and a sharp drop in Cu-N/O peak (coordination number dropped from 3.8 to 0.6) during the CO₂R process, indicating the Cu agglomeration (**Fig. 3b and supplementary Fig. 18, 19, Table S2**). This Cu agglomeration leads to decline of CH₄ selectivity during CO₂R (**Supplementary Fig. 6**)³⁰. With EDTA decoration, the Cu-Cu peak increased and Cu-N/O peak declined in the initial 30 min, then remained stable for the rest of the process (**Supplementary Fig. 20**), demonstrating the regulation of Cu ions via the chelating effect (**Fig. 3c**). We obtained small Cu clusters decorated with additional Cu-N/O sites. The fitted Cu-Cu bond coordination number of the EDTA decorated sample is smaller (5.4) than that of pristine CuPc/CNP (6.7), demonstrating the multidentate chelation constraining effect on Cu cluster size (**Supplementary Table S2**) The fitted Cu-N/O coordination number of the EDTA decorated sample was larger (2.5, **Supplementary Fig. 21, Table S2**) than the sample without EDTA decoration (0.6). We attributed the enhanced and maintained CH₄ FE (**Supplementary Fig. 11**) to the EDTA chelating effect on Cu ions -that confined Cu cluster size and generated additional Cu-N/O active sites^{13,35}”

- In new **Figure 2d** of the revised manuscript, we have presented the new stability test result.

Figure 2d Stability test of CO₂ methanation during 5 h of electrolysis under the current density of 100 mA cm⁻².

Comment #2:

In the process of electrochemical CO₂RR, pH is an important factor. The CO₂RR performance comparison of CuPc in acidic MEA and neutral MEA system also proves it in this work. The result show that the CO₂RR performance is better in neutral MEA than that the acidic MEA system. The author should provide the CO₂RR performance of EDTA/CuPc/CNP in neutral and alkali MEA system to show the superiority of acidic MEA system and explain the reason of this phenomenon.

Response #2:

We have now compared the EDTA/CuPc/CNP samples in acidic, neutral and alkaline systems. The hydrogen evolution reaction (HER) in the acidic system is similar to that of the neutral and alkaline systems (new **Supplementary Fig. 15**). The CH₄ selectivity is slightly (64 % in alkaline and 66% in neutral system) lower and the C₂H₄ is (9% in alkaline and 6% in neutral system) higher in alkaline/neutral systems, attributed to the higher cathodic pH that promotes C₂₊ products (*Nat. Commun.* 2019, 10, 32; *Science* 2018, 360, 783–787). The full cell voltage of the acidic system is similar to the neutral system at the same current density.

We then compared the CO₂ single pass conversion (SPC) of the three systems. With a total CO₂R FE >50%, a CO₂ SPC of 78% was achieved for the acidic system, nearly 4-fold the theoretical maximum of 20% for alkaline and neutral systems (**Fig. 2e, Supplementary Fig. 16**). In neutral/alkaline systems, the (bi)carbonates cross through the AEM, leading to the CO₂ loss. The CEM in the acidic MEA provided a locally acidic domain for CO₂ regeneration and CO₂ extraction within the cell to minimize CO₂ loss, and achieve high CO₂ single pass conversion.

Action #2:

- On Page 6, Lines 134-143 of the revised manuscript, we added the discussion as:

“We compared the CO₂R performances (**Supplementary Fig. 15**) and CO₂ single pass conversion (SPC) in our acidic system with the conventional neutral (0.5 M KHCO₃ anolyte) and alkaline systems (0.5 M KOH anolyte).”

“In neutral/alkaline systems, the (bi)carbonates cross through the AEM, leading to the CO₂ loss. The CEM in the acidic MEA provided a locally acidic domain for CO₂ regeneration within the cell and thereby minimized CO₂ loss (<3 v/v % CO₂ detected in the anode tail gas, **Supplementary Fig. 1**) and achieved high CO₂ single pass conversion.”

- On Page 22, **Supplementary Fig. 15** of the revised supplementary information, we added the CO₂R performance comparison of three different systems.

Figure S15 CO₂R performance comparison of acidic, neutral and alkaline MEA systems. The tests were performed at a constant current of 100 mA cm⁻². The cathode used was the optimized EDTA/CuPc/CNP sample.

- On Page 23, **Supplementary Fig. 16** of the revised supplementary information, we added the SPC of the neutral and alkaline MEA systems.

Figure S16 Single pass conversion of CO₂ at different flow rates. (a) In the neutral MEA system, a neutral 0.5 M KHCO₃ was used as the anolyte and (b) In the alkaline MEA system, 0.5M KOH was used as the anolyte. The anolyte flow rate was 5mL min⁻¹. The cathode and anode were separated by an anion exchange membrane. The SPC results were obtained at a constant current density of 100 mA cm⁻².

Comment #3:

It seems that all the total FE is lower than the unity (i.e., 100%), the authors should discuss this crucial point.

Response #3:

The total FE shown in Fig. 2 corresponds to the total gas products. We have now quantified our liquid products and reported in the new **Supplementary Fig. 13**. A small amount of formate, acetate, and ethanol were detected as liquid products, and when combined with the gas products the total measured FE approaches 100% at the same current density range (50-200 mA cm⁻²) in all three cases, within experimental error.

Action #3:

- On Page 6, Lines 126-128 of the revised manuscript, we have added:

“The FE of liquid products were quantified, and the total measured FE approaches 100% at the same current density range (50 to 200 mA cm⁻²) in all three cases, within experimental error. (**Supplementary Fig. 13**).”

- On Page 20, **Supplementary Fig. 13** of the revised supplementary information, we added the FE of liquid products.

Figure S13 Liquid products distribution of different samples. (a) EDTA/CuPc/CNP, (b) EDTA/CNP and (c) EDTA/CuPc at current range from 50 to 200 mA cm⁻².

Comment #4:

An isotopic experiment using ¹³CO₂ as substrate should be performed in order to prove the origin of CH₄, C₂H₄ and CO and guarantee its formation quantity.

Response #4:

We have performed control experiments to prove CO₂ as the carbon source in our experiments. We carried out reduction experiments under the Ar condition using EDTA/CuPc/CNP catalyst as the cathode (50 to 200 mA cm⁻²). Only H₂ was detected as reduction product (new **Supplementary Fig. 14**), indicating that EDTA and CNP were not the carbon source for the carbon-based products generated.

Action #4:

- On Page 6, Lines 128-131 of the revised manuscript, we have added:

“Control experiments are carried out under Ar conditions to rule out EDTA and CNP as the potential carbon sources in the production of carbon-based products. The exclusive H₂ production under such conditions indicates that EDTA and CNP were not reactive carbon sources (Supplementary Fig. 14).”

- On Page 21, Supplementary Fig. 14 of the revised supplementary information, we added the FE performances as function of different current densities under Ar condition.

Figure S14 FE performance under Ar conditions. The samples EDTA/CuPc/CNP was used as cathode and reduction reaction was performed at different current densities ranging from 50 to 200 mA cm⁻². The Ar flow rate was 20 sccm cm⁻².

Reviewer #2 (Comments for the Author):

In this manuscript, Fan et al. reported an EDTA/CuPc/CNP hybrid electrocatalyst for CO₂ reduction to methane with both high energy and carbon efficiency. Such catalyst could exhibit CH₄ Faradaic efficiency (FE) up to 71% at 100 mA cm⁻² with <3% loss in total input CO₂. The good performance was attributed to the hexadentate donor sites in EDTA, enabling the stabilization of Cu(II) by forming Cu-N/O single sites. Overall, it is an interesting study, but there are several technical problems, which need to be addressed before considering its publication.

Response: We thank the Reviewer #2 for the valuable comments and have acted on all points below with new experiments and analyses.

Specific Comments:

Comment #1:

The important innovation in this paper is the high single-pass CO₂ conversion efficiency of 78%. It should be noted that such high CO₂ utilization efficiency was mainly due to its operation in a zero-gap MEA comminating a cation exchange membrane (CEM) and an anion exchange membrane (AEM), which has been reported in their previous work (Joule 2022, 6 (6), 1333–1343). The result is not surprising. It should no longer be used as an innovative point in this article and spend a large space to illustrate.

Response #1:

We have shortened the description of the AEM/CEM of this acidic MEA configuration, and made it clear that the system design employed here is not the focus of the contribution.

Action #1:

➤ On Page 4, Lines 77-85 of the revised manuscript, we now write:

“**Carbon-efficient CO₂-to-CH₄ system optimization.** We integrated a cation exchange membrane (CEM) and an anion exchange membrane (AEM) combination in a zero-gap manner, as applied previously to achieve high single pass conversion in the generation of multicarbon products (**Supplementary Fig. 3**). H₂SO₄ was employed as the anolyte, providing protons to regenerate CO₂ within the MEA cell. We further incorporated various ionomers in the catalyst layer to tune the cathodic local microenvironment (local alkalinity, ion migration and CO₂ mass transport). The operating conditions and binder materials were optimized for each case and PiperIon ionomer performed best, with a moderate CH₄ FE of 25% and an H₂ FE of 45% at a current density of 100 mA cm⁻² (**Supplementary Fig. 4**).”

Comment #2:

The author emphasizes severer times that the addition of EDTA helps to suppress HER. In fact, the inhibition of H₂ mainly comes from the choice of binder and the addition of CNP. For example, by replacing the Nafion binder with Piperion, FE(H₂) dropped from 67% to 47%. Then it can farther be reduced to 20% with only CNT. The detailed role of EDTA for suppressing HER should be described.

Response #2:

We have clarified in the manuscript the role of EDTA and how – through control of cluster size and form additional Cu-N/O sites– it shifts ethylene selectivity to methane, in a way not accessible to CuPc and CNP alone (as HER results).

The role of EDTA is to constrain free Cu ions, regulating Cu cluster size and forming additional Cu-N/O sites that enhance CH₄ selectivity. With the addition of CNP, Cu clusters reduced from CuPc are well confined (*Nat. Commun.* 2021, 12, 2932). Tuning the ratio between CuPc and CNP is expected to enhance the CH₄ selectivity relative to that of C₂H₄. However, H₂ production increased with the increase in CNP sites because CNP sites are inactive for CO₂R and instead produce H₂. This competition between HER and CO₂R is more of an issue in the acidic system, where the local pH is lower. Therefore, we cannot further confine Cu cluster size to gain CH₄ selectivity through increasing CNP/CuPc ratio in acidic MEA.

EDTA provides a way to chelate Cu ions through N/O bonds, further regulating Cu cluster size and forming additional Cu-N/O sites. With the EDTA regulation, the Cu cluster size is confined to a smaller size (5.4) compared with the sample without EDTA regulation (6.7). In addition, we obtained additional Cu-N/O decoration sites when the N/O sites of EDTA bond with free Cu, facilitating the protonation steps for CO₂ to CH₄. These Cu-N/O sites decorated low-coordinated Cu shift the reaction from C-C coupling to CO₂ hydrogenation, effectively enhancing the CH₄ selectivity and minimizing HER.

Action #2:

- On Page 4, Lines 68 and 86 of the revised manuscript, we have rephrased the sentences and removed the description of “suppress HER”. We now write as:

“With EDTA decoration, we obtained low-coordinated Cu clusters decorated by Cu-N/O single sites - that facilitate CO₂R to produce CH₄.”

“To enhance the selectivity of CH₄, we deployed the low-coordination Cu strategy that is selective for CO₂ electrochemical methanation^{14,30}.”

Comment #3:

The authors believe that the formation of single Cu-O/N sites decorated low-coordinated Cu clusters is crucial for producing methane efficiently. They observed the existence of Cu-O/N sites with in-situ EXAES, but there was no direct evidence that it was the key catalytic site. Considering

previous studies that Cu clusters were also efficient in producing methane, it is confusing whether the key active site is Cu-O/N site or low-coordinated Cu cluster.

Response #3:

Cu clusters are efficient in producing CH₄, and the CH₄ selectivity is sensitive to the cluster size (*Nat. Commun.* 2018, 9, 415; *Nat. Commun.* 2021, 12, 2932). The bare CuPc/CNP sample shows a higher Cu-Cu coordination number (CN) of 6.7 that shifts the reaction from CO₂ hydrogenation to C-C coupling during the CO₂R process (**Supplementary Fig. 6**). With the decoration of EDTA, the EDTA/CuPc/CNP shows a lower Cu-Cu coordination number of 5.4, indicating a smaller cluster size compared with bare CuPc/CNP case. The reduced Cu cluster size shifts production from C₂H₄ to CH₄, which is consistent with the previously reported results that copper in a smaller coordination number favours methane than ethylene (*Nat. Commun.* 2021, 12, 2932).

Meanwhile, we found the Cu-N/O coordination number (2.5) in our EDTA case is much higher than that without EDTA (0.6), indicating the generation of additional Cu-N/O single sites via bonding Cu ions with EDTA. The N/O coordinated Cu sites are reported to be effective on enhancing CH₄ selectivity (*Nat. Commun.* 2021, 12, 586). Through our DFT calculations, we found that CuEDTA structure exhibits the lowest reaction energy for the protonation of CO₂ to *COOH. Meanwhile, we observed a thermal-neutral protonation step (step 5) - further facilitating the subsequent protonation steps for CO₂ to CH₄, providing the lowest potential determining step. Thus, we believe that Cu-N/O single sites further promote CH₄ selectivity.

Our strategy of constraining free Cu ions resulted in low-coordinated Cu and additional Cu-N/O single sites simultaneously. By chelating with EDTA, the combination of low-coordinated Cu and additional Cu-N/O decoration sites result in boosting the CH₄ selectivity.

Action #3:

- On Page 7, Lines 158-173 of the revised manuscript, we have discussed the connections between the time-dependent *in-situ* XAS results and the performance change. We have modified our description accordingly.

Comment #4:

High CH₄ selectivity has also been reported by adding chelating ligand molecules directly to the surface of polycrystalline copper (EDTMPA on commercial polycrystalline Cu, *Nat. Commun.* 2022, 13 (1), 3158.) . The authors need to compare previous reports to illustrate the major innovation of this work.

Response #4:

The previous paper (*Nat. Commun.* 2022, 13 (1), 3158) uses EDTMPA as a selective surface-capping additive onto polycrystalline Cu to selectively generate Cu (110) facets for CO₂ methanation. They focused on controlling the surface structure in promoting CH₄ production rate

in an alkaline system. The EDTMPA in this work serves only as the capping agent affecting on polycrystalline metal Cu, but not chelates with Cu(II) to generate additional active sites.

The acidic conditions considered in our work – motivated by reactant loss suffered in alkaline systems – present a unique set of challenges. Under more acidic conditions, Cu leaches out and further agglomerates. The EDTA chelating strategy is designed to address this in acidic electrolyzers, constraining Cu ions and confining Cu agglomeration. The N and O sites of EDTA coordinate with free Cu ions, regulating Cu cluster size and forming additional Cu-N/O sites. These Cu-N/O sites further promote CH₄ selectivity. The noted EDTMPA work used a surface construction strategy on a metal Cu(0) surface and the EDTMPA was used as a capping agent to obtain specific surface facet. In our case, we start with Cu(II)Pc, in which the Cu(II) leached out rapidly in acidic conditions. A chelating strategy is deployed to confine these free Cu(II) ions, obtaining low-coordinated Cu and forming additional Cu-N/O sites.

Action #4:

- We have discussed this and cited this paper as new reference 20 in our revised manuscript.

Comment #5:

In the theoretical calculation section in figure 4, why the charge of the catalyst in step 9 is not the same as the one in step 0?

Response #5:

In our DFT calculations, we considered the disodium EDTA to capture Cu(II) ions and form a series of complex structures of copper ethylenediaminetetraacetate. To determine the active site and reaction mechanism, we systematically examined the different degrees of protonation of copper ethylenediaminetetraacetate [C₁₀H_{14+/-n}CuN₂O₈]^{n+/-} (*n* = 0 and 1), and how these structures affect the reaction energy of CO₂ methanation. After thoroughly comparing the results, we proposed CO₂ protonation start with the adsorbing on the Cu active site in a structure with the molecular formula of [C₁₀H₁₄CuN₂O₈], in which two COO⁻ form chemical bonds with Cu²⁺. We found for the first half of CO₂ methanation, this structure exhibits the lowest reaction energy for the protonation of CO₂ to *COOH. Meanwhile, we expect a thermal-neutral protonation of [C₁₀H₁₄CuN₂O₈] and generation of [C₁₀H₁₅CuN₂O₈]⁺ (step 5), since [C₁₀H₁₅CuN₂O₈]⁺ facilitates the subsequent protonation steps for CO₂ to methane that provides the lowest potential determining step of 0.80 eV.

We have also calculated that, after methanation, the reaction energy for [C₁₀H₁₅CuN₂O₈]⁺ deprotonation and regeneration is approximately -0.02 eV (step 9 to step 0). We have added these points to the discussion in the manuscript.

Action #5:

- On page 8-9, Lines 199-217 and new **Fig. 4** of the revised manuscript, we have changed the description from $[\text{Cu(II)EDTA}]^0$ and $[\text{Cu(II)EDTA}]^+$ to $[\text{C}_{10}\text{H}_{14}\text{CuN}_2\text{O}_8]$ and $[\text{C}_{10}\text{H}_{15}\text{CuN}_2\text{O}_8]^+$.
- On page 8, Lines 207-209, we have added a discussion on the reaction energy of the regeneration of $[\text{C}_{10}\text{H}_{14}\text{CuN}_2\text{O}_8]$ from $[\text{C}_{10}\text{H}_{15}\text{CuN}_2\text{O}_8]^+$ as:

“We note that the protonation/deprotonation between $[\text{C}_{10}\text{H}_{14}\text{CuN}_2\text{O}_8]$ and $[\text{C}_{10}\text{H}_{15}\text{CuN}_2\text{O}_8]^+$ (**Fig. 4a, inset**) is a thermal-neutral step (with a free energy of 0.08 eV for Step 5 and -0.02 eV for Step 9 to Step 0).”

Figure 4 DFT calculations on CO_2 protonation to CH_4 . (a) Free energy diagram for CH_4 production on Cu active sites in the complex structures of $[\text{C}_{10}\text{H}_{14}\text{CuN}_2\text{O}_8]$ and $[\text{C}_{10}\text{H}_{15}\text{CuN}_2\text{O}_8]^+$. The inserted figures represent the protonation/deprotonation between $[\text{C}_{10}\text{H}_{14}\text{CuN}_2\text{O}_8]$ and $[\text{C}_{10}\text{H}_{15}\text{CuN}_2\text{O}_8]^+$. (b) Corresponding atomic configurations for each elementary step, including $[\text{C}_{10}\text{H}_{14}\text{CuN}_2\text{O}_8]$, $^*\text{CO}_2$, $^*\text{COOH}$, $^*\text{CO}$, $^*\text{CHO}$, $^*\text{OCH}_2$, $^*\text{OCH}_3$, $^*\text{O}$, $^*\text{OH}$, and $[\text{C}_{10}\text{H}_{15}\text{CuN}_2\text{O}_8]^+$.

Comment #6:

The authors should discuss the expense of their system with lower energy intensity, such as the use of both AEM and CEM, microfluid channels. I am also curious on the stability of the system operated at higher current densities.

Response #6:

The use of extra membrane layers and microchannels in the acidic system can lead to higher voltage and increased energy costs. However, from our previous study (*Joule* 2022, 6 (6), 1333–1343), the resistance penalties from the additional CEM and microchannels are low. The voltage drops within the channel layer and CEM is calculated to be <1% and <10% of the full cell voltage, respectively. The direct comparisons with single-membrane alkaline and neutral systems, now included in revised paper (**Supplementary Fig. 15**), support these findings, with similar full cell voltages in all cases.

Action #6:

- On Page 6, Lines 143-145 of the revised manuscript, we added the discussion as:

“The CEM and the integrated microchannels do not add significant ohmic resistance to the overall system³¹, as indicated by the comparable voltage with the neutral system (**Supplementary Fig. 15**).”

- We also performed a stability test at a higher current density of 200 mA cm⁻². In the initial hour, the methane selectivity declined from 63% to 40% and then remained stable during the rest of the test (see Figure below). At a higher reduction current density, the initial Cu leaching is faster than at lower current densities, leading to a rapid increase of Cu agglomeration accompanied by the decay of the Cu-N/O sites. The methane selectivity remains stable when the EDTA chelated Cu ions are maintained beyond this initial period.

Figure Stability test at a higher current density of 200 mA cm⁻².

Reviewer #3 (Comments for the Author):

In this manuscript, Fan et al. report a high selectivity (Faradaic efficiency of 71%) for the conversion of CO₂ to methane under acidic conditions on a dispersed Cu electrocatalyst at practical current densities (~100 mA/cm²). Importantly, they demonstrate a high single-pass conversion of CO₂ to this product under minimal loss of CO₂ to bicarbonate.

This work is of significance because the energy efficiency of the conversion of CO₂ under alkaline and neutral conditions is severely limited by the enviable loss of a significant fraction of CO₂ to bicarbonate under these conditions. Selective CO₂ reduction under acidic conditions is challenging because of the competition with the hydrogen evolution reaction. The authors demonstrate a clever strategy for stabilizing the undercoordinated Cu sites that promote CO₂-to-methane conversion.

The manuscript is well written, the data appears to be robust and supports the authors' claims. Methods and materials are described in appropriate detail. A strength of the work is the combination of technoeconomic analysis, catalyst design, careful reactor engineering, and catalyst characterization. Few articles approach a subject in such a comprehensive manner. The authors have thoroughly characterized their catalytic system with X-ray spectroscopy (Fig. 3) and DFT (Fig. 4). The characterization reveals no surprising insights (the mechanistic aspects have been mostly established in prior work and the active site structures on Cu have also been extensively described in the literature). Therefore, I estimate that the primary impact of this work on the field will be that it represents a new benchmark for CO₂-to-methane conversion under acidic conditions.

I did not find any technical weaknesses or issues with the presentation. The manuscript is publishable in its present form.

Response: We appreciate the Reviewer #3 for the positive comments.

REVIEWERS' COMMENTS

Reviewer #2 (Remarks to the Author):

I thank the authors for their responses. Besides the responses to my comments, I have also examined the comments from Referee #1 and the authors' responses. I believe the authors have provided satisfying answers to most of the comments on the original manuscript.

I still have a suggestion for revision before publication of the manuscript. In Table S1, the authors compared different CO₂-to-CH₄ systems. It should be noted that the current density of this work is substantially lower than the literature ones (100 vs 220-480 mA/cm²). In the responses, the authors showed the data of 200 mA/cm², and cell voltage increased to 4.2 V and methane selectivity decreased to 40%. Such data should be included in Table S1, which is helpful for the readers to understand the limitations of this system.

AUTHORS' REPLY TO THE REVIEWERS' COMMENTS

Title: Single-site Decorated Copper Enables Energy- and Carbon-efficient CO₂ Methanation in Acidic Conditions

Journal: Nature Communications (MS ID: NCOMMS-23-03695A)

We thank all reviewers again for their feedback of this work and for helpful recommendations.

Reviewer #2 (Comments for the Author):

I thank the authors for their responses. Besides the responses to my comments, I have also examined the comments from Referee #1 and the authors' responses. I believe the authors have provided satisfying answers to most of the comments on the original manuscript.

I still have a suggestion for revision before publication of the manuscript. In Table S1, the authors compared different CO₂-to-CH₄ systems. It should be noted that the current density of this work is substantially lower than the literature ones (100 vs 220 480 mA/cm²). In the responses, the authors showed the data of 200 mA/cm², and cell voltage increased to 4.2 V and methane selectivity decreased to 40%. Such data should be included in Table S1, which is helpful for the readers to understand the limitations of this system.

Response: We thank the Reviewer #2 for the valuable comment and have included the data of 200 mA cm⁻² in our revised Supplementary Table 1